# “Embracing the Inner Strength and Staying Strong”: Exploring Self-Care Preparedness among Nurses for Enhancing Their Psychological Well-Being against the Long-Term Effect of COVID-19 Pandemic in Brunei Darussalam

**DOI:** 10.3390/ijerph20176629

**Published:** 2023-08-23

**Authors:** Khadizah H. Abdul-Mumin, Aminol Azrin Maideen, Agong Lupat, Siti Nur-Diyana Mohd-Alipah, Roziah H. Mohammad-Alli, Hajah-Noraini H. Abd-Manaf, Abd-Rani Osman, Haji Mohd Reduan Abd. Fata, Nani Busrah, Cynthia Darling-Fisher, Deeni Rudita Idris

**Affiliations:** 1Pengiran Anak Puteri Rashidah Sa’adatul Bolkiah Institute of Health Sciences, Universiti Brunei Darussalam, Gadong BE1410, Brunei; aminol.maideen@moh.gov.bn (A.A.M.); agong.lupat@ubd.edu.bn (A.L.); deeni.idris@ubd.edu.bn (D.R.I.); 2School of Nursing and Midwifery, La Trobe University, Bundoora, Melbourne, VIC 3086, Australia; 3Department of Nursing Services, Ministry of Health, Bandar Seri Begawan BB3910, Brunei; roziah.alli@moh.gov.bn (R.H.M.-A.); noraini.manaf@moh.gov.bn (H.-N.H.A.-M.); rani.osman@moh.gov.bn (A.-R.O.); reduan.fata@moh.gov.bn (H.M.R.A.F.); nani.busrah@moh.gov.bn (N.B.); 4School of Tropical Medicine and Global Health, Nagasaki University, Nagasaki 852-8523, Japan; jj20220005@ms.nagasaki-u.ac.jp; 5School of Nursing, University of Michigan, Ann Arbor, MI 48109, USA; darfish@umich.edu

**Keywords:** COVID-19, environmental, health, pandemic, psychological, self-care, Brunei

## Abstract

During three years of the unprecedented, massive COVID-19 pandemic that affected the world, nurse front liners faced substantial challenges and experienced long-term adverse mental health. This study explored psychological self-care practices undertaken by nurses to strengthen their mental health and develop resilience in their professional care role while addressing the challenges of the COVID-19 era. A qualitative descriptive exploratory study was conducted on 40 nurses who worked in providing COVID-19 care across Brunei using focus groups aided by semi-structured open-ended questions. Data were thematically analyzed using constructivist grounded theory. Four themes emerged: (1) ‘Care of the mind, heart and soul’; (2) ‘Physical care for the psychological well-being’; (3) ‘Venting out and distraction’; (4) ‘Have faith, think and stay positive’. The challenges of dealing with a worldwide pandemic unintentionally placed nurses’ mental well-being at risk while the government strategized and prioritized containing and preventing the spread of and death from COVID-19. Health administrators, as well as nursing educators, need to promote and develop resources to strengthen nurses’ psychological self-care practices. This will not only benefit individual nurses but will help promote the well-being of patients and employees, improve the health of all, and counteract any unintended stressful situations, even beyond the COVID-19 pandemic.

## 1. Introduction

In the past three years, the coronavirus disease (2019) (COVID-19) and its variants have affected the world through multiple waves at different scales and intensities. The World Health Organization (WHO) [1] urged countries to take all preventive measures possible to limit virus transmission, including continuous surveillance, quarantine, awareness campaigns, and early detection. After three years, the number of COVID-19 cases and deaths continue to fluctuate daily, and many countries have moved on to the endemic phase. This new norm defined by the endemic phase has indirectly created the perspective that COVID-19 is seen as a recurring disease, comparable to the common flu. Consequently, and unintentionally, the serious risk of COVID-19 may not be viewed as life-threatening, and the risk of death may not be as compelling a concern as it has been over the past three years. The process of normalization of COVID-19 may place additional stress on healthcare providers, in particular nurses, should the pandemic recur.

Nurses take the lead in promoting health, advocating for patients, and strengthening patients’ care [2]. In fact, nurses were among the first frontline workers to provide daily care for confirmed and suspected COVID-19 patients. There is substantial research demonstrating that nurses have experienced increased stress and psychological distress in response to the unprecedented and unpredictable nature of COVID-19 and its related protocols [2,3,4]. An Italian study found that nurses had the greatest psychological problems as a consequence of the COVID-19 outbreak [5].

Anxiety, depression, loneliness, and other mental health concerns were compounded throughout the COVID-19 pandemic. Dealing with and combatting COVID-19 amplified and significantly and adversely affected the mental health of healthcare workers. A survey conducted on 1119 healthcare workers in the United States of America found that they experienced stress (93%), anxiety (86%), frustration (77%), exhaustion and burnout (76%), and felt overwhelmed (75%). Healthcare workers also shared that they are experiencing trouble with sleep (70%), physical exhaustion (68%), and work-related dread (63%) [6]. Likewise, a cross-sectional study using a self-administered online questionnaire involving 800 nurses in Spain found that 68% had some level of depression, anxiety, insomnia, and distress, and of these, 38% had moderate or severe symptoms that highly impacted those who worked in COVID-19 hospital units and in nursing homes [7]. Similarly, an American survey that investigated nurses’ perceptions of working during the early stages of the pandemic found that more than 50% of respondents experienced symptoms of depression and anxiety and close to one-third had symptoms of post-traumatic stress disorder [8].

Notably, numerous COVID-19-related studies have identified potential contributors to mental health problems among healthcare workers. These include fear of infection for oneself and one’s loved ones, the high rates of disease transmission and fatality, and fear of the unknown regarding this disease [9,10]. The struggle to keep abreast with protocols that kept changing, especially during the early days of the pandemic, contributed to the feeling of fear and stress [11]. Reports on burnout as an occupational hazard among nurses are also evident [12]. Burnout is a psychological syndrome resulting from a prolonged response to stressors on the job [13]. There is no doubt that nurses providing COVID-19-related care over the pandemic experienced multiple stressors which could lead to burnout and decreased job satisfaction. Burnout further impacts nurses’ psychosocial well-being, interpersonal relationships, and the quality of patient care provided, which, in turn, could have an effect on patient mortality [13]. Additionally, those nurses caring for dying COVID-19 patients were found to have a heightened risk of job burnout and secondary traumatic stress syndrome [14].

Research shows that nurses use various coping strategies to manage the psychological distress of working during a pandemic. Substantial quantitative studies provide an extensive contribution to our understanding of the detrimental impact of COVID-19-related care from the nurses’ perspectives [4,15,16,17,18,19]. However, these studies have limitations in terms of providing an in-depth understanding of the quality of this impact. Additionally, current qualitative studies widely focus on the evidence of the identification of needs for general coping strategies which are undoubtedly important [2,7,10,20,21,22]. In spite of this, existing studies have not adequately addressed the depth of salient features of self-care strategies, which have been found to be the initial actions taken before any other coping mechanisms are engaged. Given these challenges, it is of fundamental importance to explore these self-care strategies and their functions in improving the psychological well-being of nurses. Following the lesson learnt from the pandemic, there are also some intensive and contentious discussions on how nurses practice self-care in managing stress and coping with a range of adversities and painful experiences and suffering during the COVID-19 pandemic [20,22,23,24]. Several strategies were outlined such as self-awareness, emotion regulation, and self-compassion [20]. The ability of a person to regulate their emotions is key to avoiding experiencing the distress of others [23]. Therefore, inadequate emotion regulation leads to becoming overwhelmed. Self-kindness in self-compassion is the ability to accept oneself and avoid self-criticism and self-judgment [20]. Self-care strategies are advocated for nurses who definitely require support and help as they make sense of the COVID-19 crisis and reshape their lives [23]. It was asserted that self-care strategies lead to developing moral resilience through self-efficacy and self-control that consequently help nurses respond positively to the distress they encountered. There is also an argument that structural support and education are needed at the organizational level in order to understand the negative effects that the COVID-19 crisis has had on nurses. It was also pointed out that nurses relied on spiritual help to refresh and bolster their faith during experiences of loss and unpredictable trauma experienced during working throughout the COVID-19 pandemic [24]. To assess nurse coping during the pandemic, we conducted a nationwide qualitative study in Brunei using focus group discussions involving a total of 75 nurses. We found that varied psychosocial coping methods were employed during the different phases of the COVID-19 pandemic along with support from family, friends, the public, and at the governmental level [3]. It is noteworthy that the nurses’ first and foremost primary coping strategy was psychological self-care. This reflects an understanding of the importance of and commitment to strengthening their mental health. The nurses described self-care activities such as getting adequate sleep, healthy eating, regular exercise, social connections, and mindfulness. It can be deduced from the study that nurses demonstrate a strong sense of responsibility, autonomy, and commitment to empowering their psychological well-being positively.

The concept of self-care and self-care deficit described by Orem [25,26] provides a framework for understanding these findings and a foundation for our current research. Self-care was also popularly advocated in health promotion, prevention, and protection as the principal concept to empower health [27,28] and largely emphasized the management of chronic diseases. In the context of the COVID-19 pandemic, self-care can be viewed as behaviors that promote and maintain physical and psychological well-being, which assist nurses in promoting self-efficacy and managing their health despite the stressors they face [29]. Poor psychological health among nurses is not only detrimental to the individual but also affects professional performance and, in turn, the quality of nursing care to patients [21]. Study findings from previous pandemics such as H1N1 emphasized the importance of studying the experiences of frontline nurses to inform effective workplace and national responses during future healthcare crises [30,31]. This is essential to support organizational and workplace efforts to allay the stress and physical and mental health issues that nurses have experienced during, and in the wake of, the pandemic.

At all management levels, there was no intention to put nurses’ mental health at risk. However, the past three years of the COVID-19 era have taught us a great deal, including that healthcare systems across the globe are confronted with various shortcomings in providing care for COVID-19 patients, the efforts to contain the spread of COVID-19, prevent death [21], and normalize life amid the COVID-19 invasion. Adverse mental health was evident not only among the public but also equally and intensely felt by nurses [8]. There was a lack of preparedness for the pandemic due to the unprecedented nature of COVID-19 [32], and the lack of knowledge, skills, expertise, and equipment that consequently produced fear, anxiety, and stress [2]. This indirectly placed the nurses in a situation that required them to prioritize healthcare provision and disregard their own adverse mental health experiences. Previous studies highlighting the importance of self-care practices among nurses emphasize that nurses must have strong determination, be empowered, and be autonomously able to take charge of their mental health [3,20].

Although self-care practices among nurses were addressed in previous studies, there was not sufficient in-depth detailed exploration of the self-care practices nurses used to positively function to promote their psychological care [3,33]. Hence, the present study aimed to explore positive psychological self-care among nurses who provided COVID-19-related care and examine how these practices were adopted by nurses to strengthen their mental well-being, which enabled them to continue to function and perform their roles efficiently beyond the COVID-19 epidemic and pandemic to the endemic era and any other future crises. Therefore, the research question for this study was: “What did the nurses who provided COVID-19-related care do as their self-care strategies to positively strengthen their psychological well-being?”.

## 2. Materials and Methods

### 2.1. Ethics

The study protocol was designed and performed according to the principles of undertaking human research as documented in the Declaration of Helsinki [34]. Ethical clearance was provided by the joint committee of the Pengiran Anak Puteri Rashidah, Institute of Health Sciences Research Ethics Committee (IHSREC), Universiti Brunei Darussalam and Medical and Health Research and Ethics Committee (MHREC), and Ministry of Health (ERN: UBD/PAPRSBIHSREC/2O20/49).

### 2.2. Study Design

An exploratory qualitative approach was employed. This design was relevant for acquiring in-depth insight [35,36] into the psychological self-care of nurses who provided COVID-19-related care during the pandemic, requiring exploration to understand the phenomenon. This approach offers in-depth and detailed explorations of attitudes, reasoning, and conversational phenomena, such as focus group interviews. They are often exploratory and produce rich and detailed data that can provide essential insight into areas with little existing knowledge [37]. Exploratory qualitative research allows the participant to talk in more depth, expressing their experiences in their own words [38]. This helps researchers develop a real sense of a person’s understanding of a situation. This could not be achieved by quantitative methods, which can be restrictive with regard to certain important aspects of subjects or phenomena under study and may overlook complex issues, which are, for instance, considered significant in qualitative research [39]. The consolidated criteria for reporting qualitative research (COREQ) 32-item checklist [40] guided the design of the study.

### 2.3. Study Settings and Participants’ Characteristics

All nurses working in the national health system of Brunei, having at least two years of work experience, and working in COVID-19-related care during the study period, met the criteria to participate. COVID-19-related areas denote clinical settings where nurses provided care to confirmed and suspected COVID-19 patients.

Seven FGDs were conducted with 40 nurses, comprising first-level nurses (staff nurses and nursing officers) (*n* = 25) and second-level nurses (assistant nurses) (*n* =15) who worked in various COVID-19-related areas. Their workplaces included the National Isolation Centre (General Ward, Intensive Care Unit and High Dependency Unit), influenza-like illness settings of the emergency departments, sudden acute respiratory infection centers, swab centers, flu clinics, and isolation wards of all hospitals. The majority of the nurses were female (*n* =33) and had work experience of 10 years or more (*n* =33) prior to placement in COVID-19-related areas (See Table 1). As Brunei is such a small country, individual nurses may easily be identifiable from their affiliations with their workplace. Hence, confidentiality and anonymity are protected; their characteristics are not reported with relationships to their workplaces in Table 1.

### 2.4. Methodology for Data Collection

Seven in-depth focus groups were conducted in the different COVID-19-related care areas. Focus groups allowed participants to interactively share and exchange their experiences through open discussion [41,42]. They also gave access to subjective experiences and allowed researchers to explore intricate facets of human life-worlds [39].

Those in charge of the study sites were briefed by the research team on the inclusion criteria and afterwards offered to include in the study all eligible nurses. Nurses who expressed their interest to participate in the study were asked to attend at any of the allocated times for the focus groups, which were discussed, arranged, and agreed upon by those in charge. The focus groups were conducted at the designated private rooms adjacent to the study sites. The participants were free to withdraw at any time during the study prior to the manuscript publication and they did not have to provide any reasons.

There were unequal numbers of participants in each group but a minimum of three and a maximum of seven participants were set up to allow for interactions and also at the same time ensure equal contributions with minimal participants’ dominance during the focus groups. Three groups were attended by a combination of nurses from the National Isolation Centre (General Ward, Intensive Care Unit, and High Dependency Unit), influenza-like illness settings of the emergency departments, and sudden acute respiratory infection centers (Group 1—n = 5, Groups 2 and 3—n = 7 for each). Two groups were from the swab centers and flu clinics, (Groups 4 and 5—n = 5 for each) and two groups were from the isolation wards of hospitals (Group 6 = and Group 7 = 5).

The use of language that allows the participants to express and also be understood by the researcher is important as it represents both the data and the communication process by which data is generated [43]. Hence, our data were collected using focus groups (FGs) in the language commonly spoken (Malay and English languages). These provided flexibility to the nurses to communicate in the language(s) comfortable to them and interactively share, discuss, and exchange their experiences.

Key questions on psychological self-care practices while working in COVID-19-related care were asked (See Table 2).

### 2.5. Data Analysis

Data were analyzed using the step-by-step thematic analysis underpinned by the principles of constructivist grounded theory [44,45]. The FGs were transcribed verbatim and counterchecked by the second and fourth authors. Data analysis was facilitated by the central process of coding that resulted in coding and categorization. Initially, ‘open coding’ was conducted where words that sounded and felt the same were grouped together. Consequently, ‘focus coding’ proceeded where similar words constituted the same meaning were grouped together, resulting in the development of preliminary themes. The constant comparative method to data analysis was performed where data in a transcript and with other transcripts were compared to ensure that all the data were accounted for during the data analysis. All the researchers crosschecked and finalized the themes and subthemes to ensure the credibility and reliability of the findings (See Table 3).

## 3. Findings

Exemplar quotes representing the themes are provided to explain and describe the themes.

### 3.1. Theme 1: Care of the Mind, Heart, and Soul

This theme depicts the meaning and importance of self-care practices to strengthen the psychological health of nurses. Despite working in various COVID-19-related care settings, all the nurses equally explained psychological self-care practices as the care of the mind (thoughts), heart (emotions), and soul (inner self).


*“We must have the strength to care for our mind, calm our heart, and feed our soul. We need to think straight even in a panic situation, be calm even if we are worried, and control our emotions although we wanted to cry out loud”.*
(P11)

All of the nurses also elaborated that they had to settle their minds by embracing the inner strength to stay calm and reiterated the need to be in control of their emotions. They pointed out that as front liners, they were obliged to be strong and stay strong psychologically.


*“We have to embrace the inner strength, be strong, and stay strong in our mind, calm our heart, and make our soul at peace. The whole country is in chaos…trying to stop COVID-19 from mushrooming (spreading)…every single day and night attempting to prevent death from COVID-19…Time felt like our enemy…it felt that we always ran out of time. Time must not be wasted on being weak and fragile…for not being able to handle our thinking, feelings and emotions.”*
(P20)

They voiced that at times they felt agitated as they always felt that they ran out of time. They also expressed that they carried a huge responsibility to protect the country. They stressed that they had no time to be in turmoil, hence highlighting the need for psychological self-care.


*“Of course, we are nervous and worried. But there is no time for that. Every pound of the heartbeat, every second counts. Time is precious…The safety of the whole nation seems to revolve around each click of the time…Tik tok tik tok…People’s lives may be at stake in the blink of an eye. We must be strong in our heart, in our mind and in our soul.”*
(P37)

They summed up psychological self-care care as their ability to care for themselves rather than focusing on any burden and/or complaining. They highlighted the importance of being composed, thinking rationally, eliminating negative thinking and emotions, and strengthening their mental health.


*“It is important for us to take care of ourselves. We cannot resolve anything if we keep on whining. We cannot be strong if we keep on counting our weaknesses and calculating all our limitations. What matters the most is that we must be able to take care of ourselves. Not only physically, but also being able to think straight and clearly, keep control of ourselves, keep our mind calm, move away from all negative impulses…”*
(P04)

### 3.2. Theme 2: Physical Care for the Psychological Well-Being

This theme illustrated how psychological self-care practices were executed and strengthened by enhancing knowledge and practices that consequently facilitated preparedness for caring for suspected and confirmed COVID-19 patients. The first and foremost fear which was equally shared by the nurses was being infected and infecting others, which negatively affected them psychologically.


*“I wash my hands several times. I showered at work after I finished my shift duty. Then I showered again when I arrived home. Sometimes I have doubts about the function of the PPE (Personal Protective Equipment) that I wore. Does it really protect me? How do I ensure that I am fully protected? I even asked myself if all that I do are normal. All I care about is I do not want to carry the virus (COVID-19) with me and then spread it to my family. This was and still is my biggest fear…”*
(P27)

The nurses acknowledged that strengthening infection prevention and control practices (IPC) reassured them psychologically, hence ensuring slowing the spread of COVID-19 to others, especially their loved ones. At the early outbreak of COVID-19, they requested PPE refresher training and other IPC that are principally required to work in COVID-19-related areas.


*“Over 30 years of my work experience, I never encountered a situation as abrupt as COVID-19. It is not that we totally don’t know (how to wear PPE). We have the knowledge and have been taught the skills, but because it is not a common daily practice, we might not remember all or even forget a few of the important steps of prevention and control of transmission of communicable diseases, moreover steps for wearing them (PPE). It’s like even if you knew things to do but if you do not do them daily, then you will take them for granted and eventually forget about them. I am one of those that need the PPE training. It is peace of mind, calms my heart and strengthens me mentally… Not only to prevent me against COVID-19 but also my family.”*
(P10)

Nurses who worked in direct contact with confirmed COVID-19 patients shared that they regularly updated their knowledge and practices by searching for current evidence and learning from experiences of other countries. They stated that by doing so they were preparing themselves to be strong mentally, hence enhancing their preparedness in working in COVID-19-related care.


*“We do our homework at home. We ‘googled’ the internet (search the internet using the Google search engine) and regularly follow authentic and reliable resources such as the WHO (World Health Organization) website, CDC (Centers for Disease Control of the United States of America), and The Global Infection Prevention Control (GIPC) Network. We looked for daily updates to ensure that we keep pace with the exceptionally quick and sometimes daily changes in practices. Doing all of these strengthens our psychological health. We must be prepared mentally.”*
(P29)

Reflecting on and learning from experiences was also highlighted by the nurses as one of the physical means of strengthening their mental health. One of the ways they performed reflective practice was by thinking through what they did daily, analyzing their strengths and improving their limitations. They asserted that they also endeavored to improve their practice daily, which made them more confident and better prepared for providing care in COVID-19-related areas.


*“We were used to doing reflective journal assignments during our nursing training (educational preparation to become a nurse). I revert back to that (writing reflective journals) ever since I was placed in NIC (National Isolation Centre). I took the time to write down my experiences whenever I have the time to write, be it at work, in the car before driving home or at home. At home, I took the time to read what I have written, analysed my strengths and weaknesses, and evaluate all that I have done and whatnot. I searched the internet for research or best practices from other countries on things that I need improvement. In this way, I learnt to improve myself through my experiences and keep on improving myself through practising what I have learnt, ‘practices make perfect’, right?”*
(P16)

### 3.3. Theme 3: Venting out and Distraction

Under this theme, the nurses discussed the measures they carried out to release and relieve stress in order for them continually be strong psychologically. The most common psychological self-care practices were talking to each other about their experiences as a way of approving and disapproving of their emotions, thoughts, and practices.


*“When we have time, as long as we are away from the patients, we will vent out our feelings, usually our favorite place is the pantry. We pour what we felt, what we have done, what went wrong, and how to improve ourselves in the future. We usually talk it (the experience) out and discuss it with each other. Whatever we discussed stay within that four walls as far as possible. It helps us to drain out all that we thought and felt rather than keeping it to ourselves.”*
(P25)

Some nurses also kept a diary of their experiences by writing in a book or in the software application of their mobile phones. They considered this as a way of letting go of their emotions and thoughts, and they said that they felt better after they wrote even if what was written was not read by anybody.


*“I wrote all my experiences in a small book, like a diary. Especially my anger and frustration. It helps me to release my tension of working under constant pressure (during the COVID-19 pandemic. I don’t need anybody to read it but I don’t mind if someone read it too.”*
(P13)


*“I pour all the things in my heart and my mind to ease my soul in the ‘Notes’ app (mobile software application). It’s not good to keep them to myself. I don’t care how I wrote it (experience), the words that I used to write it, and for whatever reasons that I wrote it. It’s me and the ‘Notes’ app. makes me feel better after I let out all those negativities. It is okay…I do not need anyone to read it and tell me that what I did was right or wrong. I just need to let it (experience) out.”*
(P31)

Likewise, a few nurses used media such as Facebook and Instagram as platforms for voicing their daily experiences and emotions, albeit with limitations due to work confidentiality, professionalism, and nurses being role models during the pandemic.


*“I spill out my heart on Facebook and Instagram. Like writing a diary of my daily experiences. But I have to be mindful too… I have to maintain confidentiality and professionalism as a nurse. Nurses have become important role models during this pandemic. I have to be selective and cannot pour all of my heart there. I cannot just swear, used bad words and say things irrationally. There are audiences out there. I must be careful.”*
(P36)

A few other nurses also talked to the person closest to them such as their husbands or family members or best friends, whom they denoted as a ‘human diary’. The nurses said that the concept of a ‘human diary’ lent them listening ears, understanding of their situation, and acknowledged their emotions.


*“I just need listening ears, a shoulder to cry on, a mind that understands me and a heart that can feel what I felt. A ‘human diary’, that allowed me to pour my heart out without limitations. I don’t need sympathy. Just be there by my side, listen to me and be tolerant. It feels good every time after I let everything (whatever is related to daily experiences) out of my chest. I become positive again and save me from adverse mental health. For that, I am grateful I have my best friend who is my human diary.”*
(P12)

Aside from the above, healthy, and positive distractions were employed by nurses to relieve negative thoughts arising from negative experiences. Some of these included hiking, jogging, running, walking, and cycling. According to the nurses, focusing on the exercises deviated their attention from their anxiousness, hence strengthening their mental well-being while also maintaining their health physically.


*“Exercising is a healthy practice. Doing exercises such as hiking, jogging, running, walking, and cycling help us to distract our minds from all the bad experiences and help us to forget about our worries and stress. Our mental health was strengthened in this way.”*
(P07)

### 3.4. Theme 4: ‘Have Faith, and Think and Stay Positive’

This theme represents the nurses’ religious, spiritual, and psychosocial self-care practices. All the nurses equally shared the importance of having faith in a higher being or supreme entity such as God for protection and the positive endurance of the daily challenges that they faced. Regardless of the different religions, all the nurses were also in agreement and believed that COVID-19 was a test from God or a supreme being for their patience. Although it took time for them, they gradually were in acceptance of the test and believed that this was ‘a blessing in disguise’.


*“COVID-19 is a very new virus. Nothing much is known about it. Of course, there are dissatisfaction, people angry at you, stressors and pressures at work, I just go with the flow. The way that I stay strong is by reciting prayer, that’s my faith. I always talk to ‘Allah’ that I always remind myself that COVID-19 is a test of our patience and I always believe that it happens for a reason.”*
(P05)


*“Of course, nursing is a stressful job, but not all the time. During the pandemic, it is the other way around. It is not easy to get used to working in a situation where stress is there all the time. Time after time, eventually I become more accepting of this test (COVID-19). Although I am a freethinker, I believed in faith. When all hopes seem gone, there’s always a silver lining.”*
(P40)

A large number of nurses acknowledged that prayer in terms of invocation, supplication or requesting, or even asking for help or assistance from the supreme being strengthened their mental health. They believed that their prayers would definitely be answered and that they were protected.


*“Every night before sleep, with my children and my husbands, altogether we pray to God. We know He is always there protecting us. We would not be left alone. Jesus and Blessed Virgin Mary protect us all.”*
(P22)


*“Just by holding the Bible soothe my heart. What more if I read it? I search for spiritual words of wisdom in the Bible that help me through during the challenging time. I know I will be protected.”*
(P35)


*“I make du’a to ‘Allah’. Ask forgiveness for all our wrongdoings that we as a country might not realise that we have done wrong. I feel strong and close to ‘Allah’ every time I make du’a. I know he will answer my du’a. He (‘Allah’) promised that in the Qur’an. Maybe not immediately but gradually and definitely.”*
(P01)

More than three-quarters of the nurses who were Muslim felt that they were closer to Allah when consistently practising their ritual prayer. They deemed this action to strengthen them emotionally and mentally by conveying their hopes through prayer.


*“We keep on praying to ‘Allah’, even though we did it (praying) in our PPE during our shift duty. It (praying) is a way to communicate to ‘Allah’ that we are hopeful that this (COVID-19) will end soon. ‘Allah’ hear us, listen to us and will grant our wishes, albeit may take time, but for sure, definitely.”*
(P33)

Many nurses resorted to spiritual practices such as meditation, with yoga given as an example. They also listened to music. They asserted that the spiritual practices brought peace to them emotionally and calmed them from the chaos brought about by COVID-19.


*“It is not easy to get a decent day off. Whenever I can, I perform meditation through Yoga. It (Yoga) helps to bring peace to my mind. It made me strong. I was able to control my emotion, have faith, think positive and stay positive.”*
(P02)


*“I listen to music that can calm me. It soothed my heart, made my mind peaceful, and I felt like it (calming music) took away all my stress and bring positive energy to my body.”*
(P18)

## 4. Discussion

The current study of psychological self-care expanded on previous studies on self-care among nurses during the COVID-19 pandemic [3,46]. Aside from practicing physical care to strengthen mental health, our study fundamentally established that self-care is firstly and principally psychologically driven. For self-care to be functional in strengthening mental health, the activities undertaken should be positive in nature. It was found that central to psychological self-care is the understanding and commitment to strengthening mental health. Therefore, it is important to encourage nurses to undertake practices and activities such as physical care, emotional care, and social care, which include religious and spiritual care to promote their mental health. Uniquely, we also found that the nurses’ strong determination to embrace self-care was inherent to them as nurses and was enriched by their foundational knowledge of the underpinnings of health.

Working during an epidemic/pandemic can be physically, emotionally, and morally demanding for nurses. Self-care allows them to preserve their mental health by overcoming and adapting to stressful situations [47,48]. If consistently and positively practiced, it will eventually develop resilience, a concept widely studied in nursing [49,50]. Self-care is defined as:


*“Proactive, holistic, and personalized approach to the promotion of health and wellbeing through a variety of strategies, in both personal and professional settings, to enhance capacity for care of patients and their families.”*
[51]

The existing body of knowledge largely advocated physical care as the key to self-care with an emphasis on disease management for individuals with chronic or long-term conditions [28]. In our study, self-care is evident to be broad in scope that can be embraced, even, for the protection, promotion and maintenance, hence, preventing adverse mental health. The study findings also pointed out that mental and physical health is interrelated, inter-connected and interdependent with each other. Indeed, health is incomplete without mental health, and mental health may not be achieved without being physically healthy. It is also evident in this study that mental health includes emotional, psychological, and social well-being. Good mental health encompasses the ability to handle stress, relate to others, and make healthy choices [27]. Adverse mental health may affect individual capacities to maintain physical health not only results in mental disorders but also increases the risk for communicable and non-communicable diseases and contributes to unintentional and intentional injury [52]. If self-care focuses on negative approaches, psychological distress is likely to occur [21].

Daily self-care routines to meet our basic needs include nutrition, hygiene, exercise [26] and psychological self-care to keep anxiety at bay. A salient feature in our study is that the nurses employed physical activities as a distraction, which they found useful for keeping their sanity. The psychological wellness of the individual is one of the health benefits of physical activity and exercise [28,53]. Exercise releases endorphins and serotonin that improve mood. Our study supported previous research and systematic reviews that physical activity is positively associated with reduced occurrence of depression/anxiety disorders and poor physical health outcomes [54,55,56,57]. Individuals who do regular physical activity were found to be less unhappy compared to those who did not exercise during COVID-19 [7,58].

Another important feature of psychological self-care is “voicing out” anxiousness and stress to prevent and eliminate the negative energy to be mounting. The present study supported a study conducted in the United States of America that found nurses used journaling, gathering virtually with friends, and talking to their loved ones as a way of self-care during the pandemic [2]. This explains the importance of sharing feelings or ‘venting out’ and not just holding back emotions, keeping silent, and suffering alone. These self-care approaches allowed nurses to go back to work each day to continue to care for as best as they can for their patients throughout the pandemic. This also highlights the importance of social support and remaining in contact with friends and families during the pandemic. It is accepted that we need to practice ‘physical distancing’ during COVID-19, but not ‘social distancing’ [20]. Our study findings indicated that kind/compassionate self-talk was practiced through journaling and keeping a diary, which indicated the expression and handling of emotions responsibly [3]. Incorporation of such methods of self-care is a way to mediate self-compassion [22]. This lessened nurses’ vulnerability to caregiving fatigue, hence, improving well-being and resilience [20]. Evidence from the pre-pandemic research has also shown that resilient individuals when confronted with crises or distressing situations are less likely to experience stress and loneliness than their counterparts [59].

This study demonstrated that nurses recognized the importance of equipping themselves with knowledge and skills and showed concern about the appropriate technique of donning and doffing PPE. It is natural that nurses were anxious, especially during the early phase of the pandemic. This finding corroborates findings from a previous study in China that in the early stages of the pandemic, nurses’ lack of knowledge about the virus and fear of infecting their families caused them anxiety [48]. Worldwide, at the beginning of the pandemic COVID-19, PPE guidelines were inconsistent and ambiguous, fluctuated over time, and differed across organizations and countries [60]. Appropriate training in PPE usage is critical to ensure maximum protection, prevent cross-contamination, and reduce the risk of COVID infection in healthcare workers [61]. Keeping abreast of evidence-based practice with the appropriate knowledge and skills would lessen the anxiety. Information-seeking could help individuals identify essential information to educate themselves about the pandemic and other infection control measures, thereby reducing their stress [59].

This study also highlighted that religious and spiritual practices were an important component of self-care and should be a part of a holistic approach to positively strengthen mental health. Our study demonstrated how religious/spiritual beliefs assisted the nurses in dealing with everyday life working as a front liner. This finding reinforces various studies on the significant role of religion in reducing stress and maintenance of mental health [62,63]. Hence, it is important to recognize that spirituality and religion facilitate self-reliance, hence resilience [64], and can be a protective factor for physical and mental health [65]. In conclusion, embracing the inter-play of positive physical, emotional, and spiritual/religious care as components of psychological self-care allows a more holistic approach to strengthening mental health. Therefore, this enhances the important functions underpinned in the biopsychosocial-spiritual care model [66].

## 5. Strengths and Limitations of the Study

This study provides a comprehensive insight into self-care practices and how these functions in strengthening nurses’ psychological well-being. Heterogeneity of gender, whereby the majority of the study participants were female, might have an underlying influence on the findings of this study. However, it was not the aim of this study to compare gender differences. Notwithstanding, it is recommended that future research may be conducted to thoroughly explore gender differences in experiences of self-care practices.

## 6. Implications of the Findings

Staying psychologically strong during, pre, and post-pandemic and in the endemic era is very important. The World Health Organization noted that COVID-19 is still here and warns countries globally about the complacency of embarking on the endemic phase and gradually normalizing COVID-19 [1]. The stress caused by the pandemic will still persist during the endemic era although it may decrease in intensity, but the daily work environment of nurses remains not without stress. Resilience is vital in nursing and embracing positive psychological self-care practices, in addition to physical self-care practices, revealed nurses’ efforts towards achieving resilience. It is vital to ensure that all nurses are managing their mental health, even beyond the stressful COVID-19 pandemic. Not every nurse is endowed with the ability to employ psychological self-care; some may need to be made aware, taught, and encouraged to practice these skills so that they are self-reliant and empowered in their mental health rather than solely dependent on the healthcare system for sources of support.

## 7. Conclusions

Many studies over the past three years of the COVID-19 era pinpointed that nurses’ mental health was adversely affected due to the unprecedented event, and lack of organizational preparedness to provide care during the pandemic. The supply of PPE was insufficient, there was inadequate staffing, limited knowledge, and limited skills in dealing with the disease. Substantial recommendations emphasize strategies for addressing the limitations identified, which may not be wholly fulfilled. This study highlighted that positive psychological self-care practices contributed significantly to supporting and maintaining the mental health of nurses during the chaos of long-term crises such as COVID-19. Although some of these positive psychological self-care practices are natural reflex action responses inherent in the daily routine, religious rituals, and spiritual practices, not all nurses may realize the value of, or even have the skills underlying, psychological self-care practices. Nurses should be made aware of positive psychological self-care practices and encouraged to learn and practice accordingly. This could be through special educational offerings, reading, or online programs as well as included in professional education. For health systems and organizations to function optimally, they must pay attention to the needs of their employees, in particular the nurses, who are the largest group of frontline workers. Addressing and developing resources to promote positive psychological self-care practices will not only benefit individual nurses but will help promote the well-being of patients and employees and improve the health of all.

## Figures and Tables

**Table 1 ijerph-20-06629-t001:** Sociodemographic data of participants.

Pin Code	Gender	Age Range	Work Experience	Marital Status
P01	Male	40–44	15 to 19	Married
P02	Female	45–49	20 to 24	Married
P03	Female	50–54	25 to 29	Single
P04	Female	35–39	10 to 14	Single
P05	Male	45–49	20 to 24	Married
P06	Female	50–54	25 to 29	Married
P07	Female	45–49	20 to 24	Married
P08	Female	50–54	25 to 29	Married
P09	Female	35–39	10 to 14	Married
P10	Female	55–59	≥30	Married
P11	Female	35–39	10 to 14	Married
P12	Female	20–24	≤5	Single
P13	Female	35–39	10 to 14	Married
P14	Female	40–44	15 to 19	Married
P15	Female	40–44	15 to 19	Married
P16	Female	35–39	10 to 14	Single
P17	Female	55–59	≥30	Married
P18	Female	40–44	15 to 19	Single
P19	Female	35–39	10 to 14	Married
P20	Female	35–39	10 to 14	Married
P21	Male	35–39	10 to 14	Married
P22	Male	30–34	5 to 9	Married
P23	Female	35–39	10 to 14	Married
P24	Female	55–59	≥30	Married
P25	Female	40–44	15 to 19	Married
P26	Female	25–29	≤5	Single
P27	Female	35–39	10 to 14	Married
P28	Female	50–54	25 to 29	Married
P29	Female	55–59	≥25	Married
P30	Female	30–34	5 to 9	Married
P31	Female	25–29	≤5	Single
P32	Male	30–34	5 to 9	Single
P33	Female	40–44	15 to 19	Married
P34	Female	40–44	15 to 19	Married
P35	Male	35–39	10 to 14	Married
P36	Male	20–24	≤5	Single
P37	Female	35–39	10 to 14	Married
P38	Female	50–54	25 to 29	Married
P39	Female	50–54	25 to 29	Married
P40	Female	45–49	20 to 24	Married

**Table 2 ijerph-20-06629-t002:** Key questions for the focus groups.

What do you understand by self-care practices enhancing your psychological health?
What were the psychological self-care practices that you have undertaken?
How (when and where) were the psychological self-care practices undertaken?
What was the importance of psychological self-care practices? (How did you see that psychological self-care practices were important to you?)

**Table 3 ijerph-20-06629-t003:** Themes and subthemes.

Themes	Subthemes
Theme 1: ‘Care of the mind, heart and soul’ Explanation: The meaning and importance of self-care practices to strengthen the psychological well-being among the nurses	Settling the mind Calming the heart Feeding the soul
Theme 2: Physical care for the psychological well-being Explanation: All efforts which were undertaken to strengthen knowledge and practices to be mentally prepared in caring for suspected and confirmed COVID-19 patients	Strengthening infection prevention and control practices Evidence-based practices ‘Practice makes perfect’ - Learning from experience, writing reflective journals
Theme 3: Venting out and distraction Explanation: The measures carried out to release and relieve stress	Talking to colleagues Keeping diary - Book, phone, media, ‘human diary’ Exercising - Hiking, jogging, running, walking, cycling
Theme 4: ‘Have faith, and think and stay positive’ Explanation: Having faith in a higher being or supreme entity such as God for protection and the positive endurance of the daily challenges	Religious practices - Acceptance of the COVID-19 pandemic as a challenge/test - Blessings in disguises - Recitation of prayer words - Praying in action Spiritual practices - Meditation/yoga - Listening to calming music

## Data Availability

The datasets generated and/or analyzed during the current study are not publicly available due to restrictions on intellectual property regulations of the Research Ethics Committee.

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
