# Peer review of "“Embracing the Inner Strength and Staying Strong”: Exploring Self-Care Preparedness among Nurses for Enhancing Their Psychological Well-Being against the Long-Term Effect of COVID-19 Pandemic in Brunei Darussalam"

_ijerph, 2023, doi:10.3390/ijerph20176629_

Round 1

Reviewer 1 Report

Review report

The article aims to explore in-depth the self-care practices used by nurses who provided COVID-19 related care and to understand how and if these practices are adopted by nurses to strengthen their mental health and well-being.

The article covers an interesting and urgent topic. I particularly appreciated that the authors used a qualitative methodology which, in my opinion, is capable to explore in-depth subjective meanings and perceptions. Moreover, even if literature on this topic is wide and still ongoing, most of the studies carried out quantitative investigation. According to me, the use of qualitative research represents the strength points of the paper, and I would suggest to authors to better evidence this aspect which seems a bit underestimated by the authors themselves.

The authors provided a detailed Introduction section. They appropriately described the literature on COVID-19 consequences on societal levels as well as on mental health of nurses. As mentioned before, I would stress the fact that most of the literature carried out quantitative investigations which, of course, gave a substantial contribution to our understanding on the detrimental impact of COVID-19 related cares from the nurses’ point of view but, at the same time, did not allow to in-depth understand the quality of this impact. Moreover, I believe the paper focused on a concept, that is the self-care, which belongs to the bigger category of “individual protective factors”. Therefore, I would suggest verifying and evidencing what (and if...) literature said about the protective factors which help nurses to deal with fatigue and stress.

The methodology is appropriately described and supported by the purposes of the study. However, I do not understand why the authors decided to describe participants’ characteristics within the “Findings” section. This seems “strange” since usually participants’ characteristics are presented within the section “Methods” (which I would re-name “Materials and Methods”), even more if the authors carried out a qualitative study in which participants should usually be selected on the basis of some inclusion and exclusion criteria (that the authors should also add). Therefore, I would suggest putting the sub-paragraph 3.1 Participants characteristics within the sub-paragraph 2.3. Study settings and participants (which should be re-named Study settings and participants’ characteristics). In this part, I would also suggest eliminating the percentage, since the numbers of the study are small and using percentage is not needed. Moreover, the paper does not explain in-depth the way in which the focus groups were created, such as the way in which the participants were put in groups. Please, specify all these aspects.

Additionally, in section Materials and Methods, a paragraph explaining your methodology and its characteristics as well as why the authors believe that the focus groups represent a useful methodology to explore the topic they decided to investigate is missing. I would suggest adding it to enhance the qualitative value of the paper. Therefore, I would also suggest creating a sub-paragraph entitled Methodology and Data Collection, explaining Data analysis process within a separate paragraph.

The results are interesting and well-presented as well as discussed by the authors. Limitations of the study missed and need to be added. Among these, for example, the high heterogeneity of the group of participants (i.e., gender as well as age differences) and the fact that these differences seemed to be not considered in the analysis process.

Minor comments:

Line 29-30 (Abstract): These lines explained the aim of the study and should be put at the beginning of the abstract.

Reviewer 2 Report

I would congratulate the authors for their interest and focus on the mental health and well-being of one of the most exposed professionals during the COVID-19 pandemic, namely nurse frontliners. It is also notable for a qualitative study to include such a large sample (40 nurses) to ground their findings and conclusions.

General comments:

The research design is sound scientifically by respecting the criteria for reporting qualitative research (COREQ) 32-item checklist.  However, it would be helpful to bring more details about the research procedure (promotion and access to nurses, FGs – detailed below) in order to ensure study transparency and replicability, if some researchers wished for.

Specific comments:

-        Abstract: I would suggest moving the general objective of the study stated in lines 29-31 earlier, for example after the introductory phrase, lines 20-21. It would bring more clarity from the beginning, and then the method used will be described.

-        Section 2.2. `Study design` - please write a clear research question for the qualitative study, after the research objective in lines 144-145.

-        Section 2.3. ‘Study settings and participants’ – is a section composed of 2 phrases only, which is too little for a subtitle. It would be helpful and more coherent to move the sample profile here, immediately after the eligibility criteria and describe participants characteristics. In this case findings report only on the qualitative data, as it is a qualitative study.

-        Section 2.4. ‘Data collection and Data analysis’: please be more specific about the focus groups carried out: how many FGs, how many participants in each FG (minimum and maximum), how many in English and Malay language, duration of FGs (minimum and maximum), people conducting the FGs (their training / profession, if the same or different, if online/face-to-face).

-        Table 2 ‘Themes and sub-themes’: some sub-themes are confusing, please add quotation marks to ‘practice make perfect’, ‘human diary’ to reflect participants voice (for example it was not clear to me what the subtheme - Keeping diary – human - meant before reading in the text about the ‘human diary’ and how it was explained by a participant)

-        Sections 5 Implications and 6 Conclusions include the same idea and wording twice (see lines 495-497 and 513-515). Please choose one best place to state this or reformulate in one of the sections.

Conclusions and recommendations are valuable tools for preparing and educating nurses, organizations, and health systems to deal with the long-term effects of the pandemic and the endemic phase.
